# Oviposition-Deterrent Effect of a High-Quality Natural Zeolite on the Olive Fruit Fly *Bactrocera oleae*, under Different Conditions of Temperature and Relative Humidity

**DOI:** 10.3390/insects15040256

**Published:** 2024-04-08

**Authors:** Soultana Kyriaki Kovaiou, Anastasia Kokkari, George Floros, Nikolaos Kantiranis, Nikos A. Kouloussis, Anestis A. Filippidis, Dimitrios S. Koveos

**Affiliations:** 1Laboratory of Mineralogy and Petrology, School of Geology, Aristotle University of Thessaloniki, 54 124 Thessaloniki, Greece; skovaiou@geo.auth.gr (S.K.K.); kantira@geo.auth.gr (N.K.); anestis@geo.auth.gr (A.A.F.); 2Laboratory of Applied Zoology and Parasitology, School of Agriculture, Aristotle University of Thessaloniki, 54 124 Thessaloniki, Greece; akokkari@agro.auth.gr (A.K.); florosgd@agro.auth.gr (G.F.); nikoul@agro.auth.gr (N.A.K.)

**Keywords:** olive fruit fly, physical control, oviposition-deterrent effect, natural zeolite (zeolitic rock), clinoptilolite, NU-FILM-P^®^

## Abstract

**Simple Summary:**

In the last decades, the number of available chemical pesticides for pest control has been dramatically reduced due to their possible negative effects on human health and the environment. Therefore, there is an immediate necessity to explore alternative products to chemical pesticides for pest control, with natural zeolites (zeolitic rocks) being among the potential options. The olive fruit fly, *Bactrocera oleae* Gmelin (Diptera: Tephritidae), is one of the most harmful pests of olives in Mediterranean countries and worldwide. In this study, we evaluated the oviposition-deterrent effect of a natural zeolite on the olive fruit fly under different temperatures and relative humidity conditions. The tested natural zeolite was highly effective in deterring egg laying in *B. oleae* females under all tested conditions. These findings are poised to advance our understanding of sustainable and eco-friendly strategies for pest control in olive orchards, presenting opportunities for the development of targeted interventions that align with both environmental conservation and effective agricultural practices.

**Abstract:**

In recent years, the number of available chemical pesticides has been dramatically reduced, urging the need for the discovery of alternatives to chemical pesticide products such as, among others, natural zeolites (zeolitic rocks). We determined the mineralogical and chemical composition of a specific and continuous layer of zeolitic rock sample (ZeotP) from Petrota, Evros, Greece, and evaluated its oviposition-deterrent effect on the olive fruit fly *Bactrocera oleae* Gmelin (Diptera: Terphritidae). The tested natural zeolite contained 70 wt. % clinoptilolite, 18 wt. % amorphous material, 7 wt. % feldspars, 4 wt. % cristobalite, and 1 wt. % quartz. We tested the oviposition-deterrent effect of ZeotP mixed or not with an emulsifier adjuvant, NU-FILM-P^®^, in water and applied it to the surface of olive fruits. The ZeotP oviposition-deterrent effect on the olive fly was very high under a series of tested temperatures (17 °C, 20 °C, 25 °C, and 30 °C) and RHs (23%, 33%, 55%, 75%, and 94%). In addition, the ZeotP residual deterrent effect after equable water spraying was high, like the respective effect of the pyrethroid insecticide Decis^®^ (deltamethrin). Our results may contribute to the effective control of the olive fruit fly using an alternative to chemical pesticides: natural zeolite (zeolitic rocks) products.

## 1. Introduction

The continuous efforts for a greener and more sustainable world in order to protect the environment from industrial and agricultural uses lead, among others, to the use of alternative chemical methods that are efficient and friendly to the environment and public health. In agriculture, and specifically in pest control, the available synthetic chemical products (insecticides, herbicides, and fungicides) have been significantly reduced in the last decades due to their multiple negative effects, such as soil and water contamination, toxicity to animals and humans, and inefficient pest control due to the development of resistant populations [1,2,3]. Therefore, the development and use of alternatives to chemical pesticide products, such as inert-sorbent materials (mineral- and dust-based products), could be a promising solution. In recent years, many studies have confirmed the effectiveness of inert dust, such as kaolin, for the control of plant and stored product pests [4,5]. There are limited studies, however, concerning the insecticidal effect of zeolites and their use for pest control.

Zeolites are in abundance among microporous materials worldwide and are characterized as hydrous, microporous crystalline aluminosilicate minerals of alkali and alkaline earth elements. They consist of a unique three-dimensional framework structure, with bonding silica (SiO_4_) and alumina (AlO_4_) tetrahedral units connected together with the sharing of all the oxygen atoms. They can also be defined as crystalline inorganic polymers, with their frameworks forming pores and channels containing significant contents of water molecules and exchangeable cations (calcium, potassium, sodium, etc.) [6,7]. 

Approximately 67 types of zeolites occur naturally, and more than 200 types are synthetically prepared in the laboratory. Natural zeolites are hydrothermal and of mainly volcanic origin. They can occur both in crystallized forms found in igneous and metamorphic rocks, as well as in grains of smaller diameters accumulated in sedimentary rocks [7,8]. The main rock type, which contains great amounts of zeolite minerals, is characterized as zeolitic rock. In nature, apart from the fibrous zeolites (erionite, mordenite, roggianite, mazzite, etc.), which are hazardous for animal and human health, other non-toxic types exist, such as chabazite, phillipsite, clinoptilolite, heulandite, and analcime. Among them, HEU-type zeolites (clinoptilolite–heulandite) are of great research and economic interest. They are characterized by the presence of tabular crystals (typically 1–100 µm in size) and the presence of micro/nanopores that create infinite three-dimensional frameworks of silica, alumina, and oxygen atoms in a structure of 10- and 8-membered rings of dimensions (7.5 × 3.1 Å (V = 23.25 Å^3^), 4.6 × 3.6 Å (V = 16.56 Å^3^), and 4.7 × 2.8 Å (V = 13.16 Å^3^) [7,8,9]. 

Due to their unique characteristics, such as their porosity and ion-exchange properties, as well as their abundance, zeolites have generated worldwide interest in their use in a wide range of applications. Natural zeolites have been extensively used in industry as catalysts, ion exchangers, absorbents for oil and spills, separation agents, and agents for the removal of heavy metals from wastewater [10,11,12]. Due to the International Agency for Research on Cancer (IARC) and Food and Drug Administration (FDA) classification of zeolites as non-toxic and safe for human consumption, they are extensively used for agricultural and horticultural purposes, among others, as fertilizers, slow-releasing carriers of fertilizers and insecticides, feed additives, in soil amendment, water purification, and gas absorption [13,14,15]. 

The olive fruit fly, *Bactrocera oleae* Gmelin (Diptera: Tephritidae), is the most harmful pest of olives worldwide, causing serious damage to olive production all over the world, especially in Greece and Mediterranean countries [16,17]. Mature females lay their eggs inside the olive fruit, and hatching larvae feed in the mesocarp, causing fruit damage [16,17] and olive oil quality deterioration as well as premature fruit drop [18,19,20]. In Italy, olives bearing up to 45% olive fly exit holes and harvested before the end of October to mid-November produced high-quality virgin olive oil, although late harvesting can lead to worse results at the same olive fly exit hole levels [21].

In Mediterranean regions, there may be more than five overlapping generations from late summer through the winter and early spring if fruits are present on the trees and winter field temperatures are suitable for development [22,23].

The olive fly’s behavior is strongly dependent on its host fruit and is related to various physical and chemical stimuli [24,25,26,27,28,29,30,31]. The olive fruit favors the ovarian maturation of the females due either to contact or volatile stimuli, as well as to bacteria that are present on its surface [24,25,26]. In addition, olive fruit volatile and contact chemical stimuli favor mating success and egg production in the olive fly [27,28,32,33]. Certain olive fruit organic volatile compounds, such as n-octane and α-pinene, as well as a mixture of n-octane, α-pinene, limonene, ethyl hexanol, nonanal, n-dodecane, decanal, and n-tetradecane, favored successful reproduction and oviposition of the olive fruit fly [27].

The control of the olive fly is mainly based on numerous preventive treatments, such as repeated bait or cover insecticide sprays not linked to efficient infestation thresholds, which usually results in the development of olive fly populations with high levels of resistance [34], failure of effective control, and the presence of chemical residues in olives and olive oil [20]. It is, therefore, of crucial importance to develop innovative and environmentally safe methods and products for the control of the olive fly.

Previous studies have documented the effectiveness of inert bags of dust, i.e., kaolin, diatomaceous earth (DE), and natural zeolites, against a broad spectrum of insect species [35,36,37,38]. The effectiveness of the inert dust against insects is mainly due to partially removing the insect epicuticle through abrasion by hard, non-sorptive particles or by disrupting the epicuticle via the absorption of epicuticular lipids by sorptive particles. Both processes induce rapid water loss from the insect body and cause death by dehydration [39,40,41,42]. Inert specks of dust may also be effective against fungi and bacteria that may co-infest stored products [43]. Yet DEs and related materials have insecticidal effects against various Blattodea, Coleoptera, Diptera, Hemiptera, Hymenoptera, and Lepidoptera [44].

Natural zeolites have a high insecticidal effect, when applied as a dust, against a number of insect pests that infest stored products, such as the rice weevil *Sitophilus oryzae* L. (Coleoptera: Curculionidae), the red flour beetle *Tribolium castaneaum* (Herbst) (Coleoptera: Tenebrionidae) [39], the confused flour beetle *Tribolium confusum* (Jacquelin du Val) (Coleoptera: Tenebrionidae), the sawtoothed grain beetle *Oryzaephilus surinamensis* (Linnaeus (Coleoptera: Silvanidae), and the maize weevil *Sitophilus zeamais* (Motschulsky (Coleoptera: Curculionidae) [45,46,47].

In a recent study by our research group, a high-quality Greek natural zeolite was found to have a high insecticidal effect on adults of the bean weevil *Acanthoscelides obtectus* (Say) (Coleoptera: Bruchidae) [40]. This high-quality natural zeolite contained 92 wt. % clinoptilolite, 3 wt. % mica + clay minerals, 3 wt. % quartz, and 2 wt. % feldspars and had a high insecticidal effect under various temperature and humidity conditions [40].

Among the inert specks of dust, kaolin has been extensively used commercially with the trade name Surround^®^ for the control of the olive fly deterring egg laying, as the presence of the particles of the product on the fruit surface could be an obstacle for the olive fruit recognition by the females [48,49,50,51]. In addition, kaolin has a deterrent effect on a number of other insect pests, such as the Mediterranean fruit fly *Ceratitis capitata* (Wiedemann) (Diptera: Tephritidae) [52,53], the cherry fruit fly *Rhagoletis cerasi* (Linnaeus (Diptera: Tephritidae) [54], and the spotted wing drosophila fruit fly *Drosophila suzukii* (Matsumura (Diptera: Drosophilidae) [55]. Kaolin also has some insecticide activity for other insect pests such as the leafminer *Liriomyza huidobrensis* (Blanchard) (Diptera: Agromyzidae), the green peach aphid *Myzus persicae* (Sulzer) (Hemiptera: Aphididae) [56], the boll weevil, *Anthonomus grandis* (Boheman) (Coleoptera: Curculionidae) [57], the leafhopper *Empoasca vitis* (Göthe) (Hemiptera: Cicadellidae) [58], and the cotton bollworm, *Helicoverpa armigera* (Hübner) (Lepidoptera: Noctuidae) [59]. Other minerals, such as bentonite, calcium carbonate, calcium hydroxide, calcium oxide, dolomite, sulfur, talc, and zinc oxide, may also have some insecticidal activity [60].

The use of alternatives to chemical insecticide products, such as inert specks of dust, with olive fly deterrent/repellent effects may contribute to overcoming the negative consequences of the use of pesticides and efficiently protecting olive production [16,17]. The aim of the present study is to characterize and assay the potential use of a novel Greek natural zeolite product in insect pest control. We determined the mineralogical and chemical composition of a high-quality Greek natural zeolitic rock and evaluated its oviposition-deterrent effect on the olive fruit fly *B. oleae* under different temperatures and RH conditions.

## 2. Materials and Methods

### 2.1. Stock Colony and Experimental Flies

Our laboratory colony of *B. oleae* was established with adults that emerged from field-collected olive fruits in October 2021 from the area of Chalkidiki, northern Greece. The stock colony flies were maintained in wooden cages (30 × 30 × 30 cm) with wire screen sides in a climatic room at 25 °C, relative humidity (RH) of 65%, and a 16L:8D photoperiod. The experimental flies were second-generation descendants of the stock colony. They were reared from egg to pupae in olive fruits, which were collected in early August and, since then, have been maintained at 5 ± 1 °C for 2–3 months before being offered to the stock flies for oviposition. For the experiments, stock females oviposited in olive fruit of the variety “Megaritiki.” After oviposition, the infested olives were placed in plastic cups, covered with wet hessian fabric, and kept in an incubator at a constant temperature of 20 °C, 70% relative humidity (RH), and 16L:8D photoperiod. The newly formed pupae were collected daily and placed in Petri dishes. Emerging females and males were maintained in wooden cages like those used for the stock colony at 25 °C, 70% RH, and 16L:8D photoperiod and used in the experiments when they were ten days old. Stock and experimental flies were provided ad libitum with a protein liquid diet of water, yeast hydrolysate (MP Biomedicals, LLC^®^, Solon, OH, USA), and sugar (v:w:w 5:4:1). The mortality of the flies during the experiments was very low to zero. However, when a female fly died during the experiment, she was replaced by another one of the same age from the laboratory colony.

### 2.2. Natural Zeolite Characterization

The natural zeolite sample (ZeotP) used in our experiments was collected from specific and continuous layers of zeolite-rich volcaniclastic tuffs in the surrounding area of Petrota, Evros region, northern Greece. A petrographic investigation of ZeotP was performed on a thin section by a polarized microscope. The mineralogical composition was determined by X-ray diffraction method (XRD) with a Philips PW1710 diffractometer equipped with Ni-filtered CuKα radiation on a randomly oriented sample from 3° to 63° 2θ at a scanning speed of 1.2°/min. The chemical composition of ZeotP was performed using the ICP-MS method. The cation exchange capacity (sorption ability) was measured by the ammonium acetate saturation (AMAS) method [61].

### 2.3. Oviposition-Deterrent Test Effect

A portion of ZeotP with the appropriate grain size distribution (<63 μm) was prepared for the experiments. ZeotP was pulverized in a mechanical agate mortar for approximately 10 min and then sieved to a size of <63 μm in a mechanical sieve shaker for 15 min. The procedure was repeated until the entire amount of natural zeolite was sieved and the processed dust was collected.

ZeotP dust suspended in water (5 g dust to 100 mL water) with or without the addition of the adjuvant NU-FILM-P^®^ (Pinoline 96%, Elanco Hellas S.A.), Athens, Greece. For the experiments, olive fruits were immersed in ZeotP suspensions in glass jars for 20 s or as a control in water. After immersion, the fruits were allowed to dry on the surface of a metal sieve and then were transferred to the base of a plexiglass cage (20 × 20 × 20 cm). In each cage, five females of *B. oleae* and five olives were transferred. To evaluate the oviposition-deterrent effect of the tested natural zeolites, we scored, under a stereoscope (Zeiss Stemi 305^®^), the number of oviposition holes in the olive fruits after 3 and 6 days of the olives’ exposure to the egg-laying females. In each treatment, there were four replicates (cages with olives and females). The experiments were carried out in a climatic room with a temperature of 25 °C, a 16L:8D photoperiod, and an RH ranging from 50 to 55%.

### 2.4. Effect of Temperature

The oviposition-deterrent effect of ZeotP with the adjuvant NU-FILM-P^®^ was evaluated at four constant temperatures using the same experimental procedure explained in the above paragraph. The plexiglass cages with the treated olive fruits and the experimental females were maintained in climatic incubators at a series of constant temperatures (17°, 20°, 25°, and 30 °C) with a 16L:8D photoperiod and at an RH ranging from 50 to 55%. At each tested temperature, there were four replicates, i.e., plexiglass cages with five females and five olive fruits.

### 2.5. Effect of Relative Humidity (RH)

In this series of experiments, the oviposition-deterrent effect of ZeotP was evaluated under five different relative humidity (RH) levels. For the experiments, the female flies with ZeotP-treated olives were maintained at different RH levels (23, 33, 55, 75, and 94%) at 25 °C and a 16L:8D photoperiod. To maintain the flies at different RHs, we used sealed plastic containers (30 × 40 × 30 cm) with an appropriate saturated water salt solution at their bottom (5 cm deep) [62,63]. A metal net was placed approximately 3 cm above the surface of the salt solution and served as the floor for the cages with experimental flies and olives. These pot cages were modified cylindrical cups of 430 cm^3^ volume (6.5 × 9.5 × 10 cm, top diameter × base diameter × height) with two circular openings (diameter: 2 cm) covered with nylon mesh to allow adequate ventilation. The cups were placed upside down on a petri dish bearing five equidistant circular holes (diameter: 1.5 cm) where the treated olive fruits were placed. The pot cages with the *B. oleae* females and the treated olives were maintained for 3 and 6 days at the different RH levels, and subsequently, the number of oviposition holes in the fruits was scored. The saturated salt solutions used in the experiments and the respective RHs inside the cages are shown in Table 1. In each RH level, there were four-pot cages (replicates) housing five *B. oleae* females and five olive fruits. 

### 2.6. Residual Oviposition-Deterrent Effect of ZeotP on Olives after Water Spraying

We determined the residual oviposition-deterrent effect of ZeotP after spraying the treated olives with water. The objective of this group of experiments was to investigate the washing effect on the tested products of water spraying. For the experiments, olives were immersed in ZeotP with and without the NU-FILM-P^®^ adjuvant, as described in the aforementioned paragraph. After ~1 h of the ZeotP application, the treated olives were placed in glass Petri dishes 9 cm in diameter in the basement of a Potter precision spraying tower (Buckard Manufacturing Co., Ltd., Rickmansworth, Hertfordshire, UK) calibrated at 0.68 atm pressure and sprayed with two constant quantities, 5 mL or 15 mL of water/min. As a control, we used olives treated with the insecticide Decis 25 EC (deltamethrin 2.5%, Bayer Hellas A.B.E.E., Marousi, Greece) with or without NU-FILM-P^®^ adjuvant. We selected as a chemical reference product the insecticide Decis^®^ because it is widely used for the control of the olive fly and may have a repellent effect, as already observed for mosquitoes [64]. After water spraying, the olive fruits were transferred to plexiglass cages with *B. oleae* females, and the number of oviposition holes in the olives was scored after 3, 6, and 9 days. In each treatment, there were four replicates, i.e., plexiglass cages with five females and five olive fruits. The experiments were carried out in a climatic room with a temperature of 25 °C, a 16L:8D photoperiod, and an RH ranging from 50 to 55%.

### 2.7. Statistical Analysis

In order to test the effects (main interactions) of different temperatures (17°, 20°, 25°, and 30 °C), relative humidity levels (23, 33, 55, 75, and 94%), and water spraying levels (5 mL or 15 mL of water/min) on the oviposition-deterrent effect of ZeotP on *B. oleae*, the ANOVA method was used within the methodological frame of General Linear Models (GLM). Subsequent significant differences (*p* ≤ 0.05) in means were further separated with the Student–Newman–Keul’s test. Levene’s test was used to confirm the homogeneity of variances among treatments. Kolmogorov–Smirnov’s test was used for testing the normality of the models’ residuals. In cases where heterogeneity of variances was significant, appropriate data transformation (log transformation) was conducted before the analysis. In those cases where the transformations failed to satisfy the criteria for parametric analysis, the Kruskal–Wallis non-parametric ANOVA, followed by the Mann–Whitney U test for all possible pairwise comparisons, was conducted. A *t*-test was used to compare the mean number of *B. oleae* oviposition holes between treated and untreated (control) olives within each of the different temperatures, relative humidity, and water spraying levels, once after 3, once after 6 days, and once after 9 days (for water spraying levels). In all hypothesis testing procedures, the significance level was set at a = 0.05 (*p* ≤ 0.05). All analyses were performed using IBM SPSS Statistics 24.0 (IBM Corp., Armonk, NY, USA).

## 3. Results

### 3.1. Natural Zeolite Characterization

Semi-quantitative estimates of the abundance of the mineral phases were derived from the XRD data using the intensity (counts) of certain reflections as well as the density and mass absorption coefficients for CuKa radiation of the minerals present [12]. As shown in Table 2, ZeotP consists of 70 wt. % HEU-type zeolite (clinoptilolite), 1 wt. % quartz, 4 wt. % cristobalite, 7 wt. % feldspars (K-feldspars and plagioclase), and 18 wt. % amorphous material. The natural zeolite sample can be characterized as clinoptilolite zeolitic rock. The cation exchange capacity (sorption ability) was measured at 193 meq/100 g.

Regarding the chemical composition (Table 3), ZeotP contains 68.43 wt. % SiO_2_ and 11.60 wt. % Al_2_O_3_. The content of CaO is 3.80 wt. %, K_2_O is 2.12 wt. %, whereas MgO content is 0.96 wt. %. Low amounts of Na_2_O, Fe2O3tot, and MnO were measured.

The mineralogical composition aligns both with the chemical composition and cation exchange capacity. Except for SiO_2_ and Al_2_O_3_, ZeotP contains sufficient amounts of CaO and K_2_O (Table 3). 

### 3.2. ZeotP Oviposition-Deterrent Effect

The application of an aqueous solution of ZeotP on olives, with and without NU-FILM-P^®^, resulted in a significant decrease in the number of oviposition holes in them. As shown in Table 4, when the female flies had access to olive fruits for 3 and 6 days, a mean number of 107.0 and 168.5 oviposition holes, respectively, were scored in the non-treated olives, whereas no oviposition holes were scored in olives treated with a mixture of ZeotP and NU-FILM-P^®^. An intermediate mean number of 16.2 and 59.0 oviposition holes were scored in olives sprayed with ZeotP without NU-FILM-P^®^. These results show that the addition of NU-FILM-P^®^ adjuvant significantly increased the oviposition-deterrent effect of ZeotP.

### 3.3. Effect of Temperature

As shown in Table 5, at all the tested temperatures, there was a significant oviposition-deterrent effect of ZeotP (with or without the addition of NU-FILM-P^®^). The number of oviposition holes in olives after the application of ZeotP with or without NU-FILM-P^®^ was very low to zero in all the tested temperatures. In a few cases in the control, there was a large variability in the number of oviposition holes, which may be due, among other reasons, to the different size and maturation status of the field-collected olive fruits used in the experiments. These results show that irrespective of temperature conditions, the oviposition-deterrent effect of ZeotP is high, in particular, when applied in combination with the adjuvant NU-FILM-P^®^.

### 3.4. Effect of Relative Humidity (RH)

As shown in Table 6, in all RHs tested, there was a significant oviposition-deterrent effect of ZeotP. The number of oviposition holes in olives after the application of ZeotP was significantly lower compared to the control at all levels of relative humidity (23, 33, 55, 75, and 94% RH).

### 3.5. Residual Oviposition-Deterrent Effect of ZeotP on Olives after Water Spraying

As shown in Table 7, after spraying with water the ZeotP-treated olives, the oviposition-deterrent effect of ZeotP, with or without NU-FILM-P^®^, remained high, and the number of oviposition holes in the fruits was significantly lower than the respective number of oviposition holes in the control (non-treated olives). However, in almost all cases, the addition of NU-FILM-P^®^ in ZeotP suspension resulted in a higher residual oviposition-deterrent effect after water spraying. The residual ZeotP oviposition-deterrent effect after water spraying was similar to the respective effect of the pyrethroid insecticide Decis^®^. 

## 4. Discussion and Conclusions

Inert specks of dust such as kaolin, diatomaceous earth (DE), and natural zeolites have an insecticidal efficacy that is mainly due to disruption of the insect’s outer cuticle (epicuticule) [35,36,37,38,39,40,41,42] or to a deterrent/repellent effect [48,49,50,51,52,53,54,55]. The physical properties of natural zeolites, such as particle size, shape, surface area, etc., as well as the chemical and thermal stability that depends on the Si/Al ratios (i.e., the least stable natural zeolites have low Si/Al ratios), affect their activity [40]. Here, we used a high-quality natural zeolite (ZeotP) with unique characteristics such as a high content of clinoptilolite and microporous minerals, a high cation exchange capacity, and a content of K and Ca. The XRD analysis of ZeotP has shown that quartz is in very low quantities (1 wt. %), and fibrous minerals are completely absent, which makes ZeotP safe for human health.

Zeolites are highly toxic against a number of insect pests that infest stored products, such as *S. oryzae* and *T. castaneaum* [46,47], *T. confusum* and *O. surinamensis* [45], *S. zeamais* [45,47], and bean weevils *A. obtectus* [40]. Our results show that ZeotP, after appropriate processing and application in the form of aqueous suspension on the surface of the olive fruit, can significantly prevent the egg laying of the olive fruit fly and, therefore, protect the olives from injury. In addition, the adjuvant NU-FILM-P^®^, mixed with ZeotP, significantly increases the oviposition-deterrent effect.

Previous studies by our group have shown that a high-quality Greek zeolite has a high insecticidal activity for the bruchid *A. obtectus* under different temperatures and RH conditions [40]. Here, we found that ZeotP, mixed with the adjuvant NU-FILM-P^®^ and applied to olives, has a high oviposition-deterrent effect on the olive fly under different temperature and RH conditions. This high oviposition-deterrent effect of ZeotP with NU-FILM-P^®^ is maintained even after spraying with water the ZeotP-treated olives and is similar to the respective effect of the pyrethroid insecticide Decis^®^ (deltamethrine), which is extensively used in the field against the olive fly and may have a repellent effect for the flies as it happens in mosquitoes [64].

The studied high-quality zeolitic rock (ZeotP), rich in HEU-type zeolite, has a high oviposition-deterrent effect against the olive fly *B. oleae*, particularly in preventing females from laying their eggs on olive fruits. This prevention of oviposition may be due to the creation of a thin layer of natural zeolite on the surface of the olives, which may mechanically prevent females from adhering to the surface of the fruit and laying their eggs. The thin layer of ZeotP formed on the olive surface prevents the oviposition process and fruit injury at a range of different temperatures and RH levels. The combination of ZeotP with an adjuvant (NU-FILM-P^®^) significantly increased the oviposition-deterrent effect and is a promising approach for olive fly control. As it occurs with other adjuvants [65], NU-FILM-P^®^ may alter the physical properties of the application, improving adhesion and emulsification and, therefore, enhancing the oviposition-deterrent effect of ZeotP. 

The oviposition-deterrent effect of ZeotP may also be due to changes in the emitted fruit’s volatile [27,28] and color [29], which have a significant effect on egg production and the behavior of the olive fly.

Our natural zeolite (ZeotP) has a similar mode of action to Kaolin (Surround^®^), which has been used for the control of the olive fly [48,51,52] and other fruit flies [52,60,66,67]. Kaolin, as our ZeotP, prevents either the landing of the flies on the olives or deters oviposition after landing due to its repellent nature, anti-ovipositional qualities, and/or its highly reflective white coating [48]. According to Salerno et al. [66], kaolin nanoparticle film significantly impacted the attachments of the Southern green stink bug *Nezara viridula* Linnaeus (Heteroptera: Pentatomidae) and the Mediterranean fruit fly *C. capitata* on chili fruits and reduced infestation. Kaolin- and copper-based compounds are viable candidates for inclusion in organic and integrated pest management initiatives. Kaolin demonstrates efficacy in reducing *B. oleae* oviposition in olive fruits [49,68,69]. Copper exhibits both a repellent effect for adults of *B. oleae* and antibacterial properties, averting the symbiotic relationship between larvae and bacterial symbionts [69,70,71]. Certain minerals, such as talc, calcium oxide, calcium hydroxide, and calcium carbonate, when applied to the surface of chili fruits, reduced the number of eggs laid on them and the infestation by the oriental fruit fly *B. dorsalis* [60].

The use of alternative pesticide products, such as our high-quality natural zeolite ZeotP with olive fly oviposition-deterrent effect, may contribute to overcoming the negative consequences of the use of chemical pesticides and efficiently protecting olive production. Field experiments with ZeotP, according to the requirements of European legislation for the specific applications, are required to confirm its oviposition-deterrent effect and efficacy for the control of the olive fly.

## Figures and Tables

**Table 1 insects-15-00256-t001:** Saturated salt solutions were used with the corresponding RHs and saturation deficit.

Solutions	Relative Humidity (%)	Saturation Deficit (kPa)
LiCl·H_2_O	23	2.96
MgCl_2_·6H_2_O	33	2.25
Mg(NO_3_)_2_·6H_2_O	55	1.51
NaCl	75	0.84
KNO_3_	94	0.17

**Table 2 insects-15-00256-t002:** Semi-quantitative mineralogical composition (wt. %) and ion exchange capacity (meq/100 g) of Petrota natural zeolite sample ZeotP.

Sample	ZeotP
HEU-type zeolite (clinoptilolite)	70
Quartz	1
Cristobalite	4
Feldspars	7
Amorphous material	18
Total	100
Ion exchange capacity (meq/100 g)	193

**Table 3 insects-15-00256-t003:** Chemical composition of Petrota natural zeolite sample ZeotP.

Sample	SiO_2_	Al_2_O_3_	Fe_2_O_3tot_	MnO	MgO	CaO	SrO	BaO	Na_2_O	K_2_O	H_2_O	Total
ZeotP	68.43	11.60	bdl *	bdl	0.96	3.80	bdl	bdl	0.03	2.12	13.03	99.97

* bdl: below detection limit.

**Table 4 insects-15-00256-t004:** Mean (±SE) number of *B. oleae* oviposition holes in olives immersed in 5% ZeotP aqueous solution, with and without NU-FILM-P^®^ adjuvant. ZeotP-treated and non-treated (control) olives were maintained in cages with five *B. oleae* females, and the number of oviposition holes was determined after 3 and 6 days at a temperature of 25 °C, a 16L:8D photoperiod, and an RH ranging from 50 to 55%.

Product	Mean (±SE) Number of Oviposition Holes after	
	3	6
	Days	
Control	107.0 ± 8.4 a	168.5 ± 12.2 a
ZeotP without NU-FILM-P^®^	16.2 ± 6.4 b	59.0 ± 15.7 b
ZeotP with NU-FILM-P^®^	0.0 ± 0.0 c	0.0 ± 0.0 c

Means within a column followed by a different letter are significantly different at the 95% confidence level (*p* < 0.05).

**Table 5 insects-15-00256-t005:** Mean (±SE) number of *B. oleae* oviposition holes in olives immersed in 5%/ZeotP aqueous solution with NU-FILM-P^®^ adjuvant at four (17, 20, 25, and 30 °C) constant temperatures. ZeotP-treated and non-treated (control) olives were maintained in cages with five *B. oleae* females, and the number of oviposition holes was determined after 3 and 6 days at a 16L:8D photoperiod and at an RH ranging from at 50–55%.

Temperature	Mean (±SE) Number of Oviposition Holes after			
	3		6	
	Days			
	Control	ZeotP with NU-FILM-P^®^	Control	ZeotP with NU-FILM-P^®^
17 °C	13.5 ± 7.7 aA	0.0 ± 0.0 aB	35.0 ± 17.0 aA	0.2 ± 0.2 aB
20 °C	9.7 ± 2.9 aA	0.0 ± 0.0 aB	37.5 ± 10.5 aA	0.0 ± 0.0 aB
25 °C	81.5 ± 30.8 bA	0.2 ± 0.2 aB	129.0 ± 28.7 bA	0.7 ± 0.7 aB
30 °C	49.2 ± 14.9 cA	0.2 ± 0.2 aB	100.7 ± 22.2 bA	1.2 ± 0.2 bB

Means within a column followed by a different lowercase letter and within a line and the same oviposition period (3 and 6 days) by a different uppercase letter are significantly different at the 95% confidence level (*p* < 0.05).

**Table 6 insects-15-00256-t006:** Mean (±SE) number of *B. oleae* oviposition holes in olives immersed in 5% ZeotP aqueous solution with NU-FILM-P^®^ adjuvant at five different RH levels (23, 33, 55, 75, and 94%). ZeotP-treated and non-treated (control) olives were maintained in cages with five *B. oleae* females, and the number of oviposition holes was determined after 3 and 6 days at a temperature of 25 °C and a 16L:8D photoperiod.

RH %	Mean (±SE) Number of Oviposition Holes after			
	3		6	
	Days			
	Control	ZeotP with NU-FILM-P^®^	Control	ZeotP with NU-FILM-P^®^
23	33.2 ± 17.0 aA	4.7 ± 2.7 aB	47.5 ± 21.4 aA	9.5 ± 0.6 aB
33	34.5 ± 10.4 aA	9.2 ± 1.1 aB	69.5 ± 8.3 aA	10.0 ± 1.1 aB
55	73.2 ± 10.2 bA	4.2 ± 1.9 aB	84.2 ± 5.6 cA	7.2 ± 3.1 aB
75	60.7 ± 8.4 bA	3.0 ± 2.3 aB	90.5 ± 2.7 cA	6.5 ± 3.1 aB
94	27.0 ± 3.3 aA	7.0 ± 1.1 aB	43.0 ± 3.4 aA	11.0 ± 1.1 aB

Means within a column followed by a different lowercase letter and within a line and the same oviposition period (3 and 6 days) by a different uppercase letter are significantly different at the 95% confidence level (*p* < 0.05).

**Table 7 insects-15-00256-t007:** Residual oviposition-deterrent effect of ZeotP after spraying with water the treated olive fruit. Olives were immersed in 5% ZeotP with or without NU-FILM-P^®^ and subsequently sprayed with 5 and 15 mL of water. Then, the olives were maintained in cages (five olives with five *B. oleae* females), and the number of oviposition holes was determined after 3, 6, and 9 days at a temperature of 25 °C, 16L:8D photoperiod, and RH ranging from at 50–55%.

Product	Water Volume (mL)	Mean (±SE) Number of Oviposition Holes after					
		3		6		9	
		Days					
		With NU-FILM-P^®^	Without NU-FILM-P^®^	With NU-FILM-P^®^	Without NU-FILM-P^®^	With NU-FILM-P^®^	Without NU-FILM-P^®^
ZeotP	0	0.0 ± 0.0 aA	7.5 ± 4.1 aB	0.0 ± 0.0 aA	39.7 ± 12.8 aB	0.0 ± 0.0 aA	56.5 ± 14.1 aB
	5	1.0 ± 0.7 aA	10.5 ± 6.1 abB	1.7 ± 0.5 aA	34.2 ± 19.9 aB	3.0 ± 1.1 aA	42.3 ± 14.6 bB
	15	0.0 ± 0.0 aA	13.8 ± 5.5 bB	0.0 ± 0.0 aA	41.8 ± 8.0 aB	1.8 ± 1.4 aA	60.8 ± 20.5 aB
Decis^®^	0	2.8 ± 0.9 aA	4.5 ± 1.8 aA	3.0 ± 1.1 abA	16.5 ± 7.0 bB	4.7 ± 2.1 aA	32.2 ± 18.0 bcB
	5	1.0 ± 1.0 aA	13.5 ± 6.2 bB	1.5 ± 1.2 aA	36.2 ± 11.6 aB	6.2 ± 0.9 aA	66.2 ± 18.7 aB
	15	3.0 ± 1.3 aA	0.8 ± 0.8 cA	4.5 ± 0.5 bA	4.5 ± 4.5 cA	4.5 ± 2.4 aA	22.5 ± 10.7 cB

Means within a column followed by a different lowercase letter and within a line and the same oviposition period (3, 6, and 9 days) by a different uppercase letter are significantly different at the 95% confidence level (*p* < 0.05).

## Data Availability

Data are contained within the article.

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
