# Peer review of "Oviposition-Deterrent Effect of a High-Quality Natural Zeolite on the Olive Fruit Fly Bactrocera oleae, under Different Conditions of Temperature and Relative Humidity"

_insects, 2024, doi:10.3390/insects15040256_

Round 1

Reviewer 1 Report

Comments and Suggestions for Authors

The oviposition deterrent effect of zeolite vs. olive Bactrocera oleae is not so surprising, considering the known efficacy of kaolin and other powders. A comparison with kaolin should be very interesting to test zeolite oviposition deterrence. The manuscript clearly demostrates zeolite efficacy, especially when added to NU-FILM-P adjuvant, and the authors correctly state in the conclusions that field trials will allow us to fully evaluate its effectiveness in olive fly control and cost-effectiveness.

Authors have to check all citations of references that in many cases are not the right ones. There is a lot of confusion about cited numbers.

Some little modifications are suggested inside the notes visible in the enclosed pdf

Comments on the Quality of English Language

Few corrections are suggested inside the notes of the attached  pdf

Reviewer 2 Report

Comments and Suggestions for Authors

The manuscript refers to the laboratory evaluation of a natural zeolite as a product to prevent olive fruit fly females from egg-laying in olives. Authors performed a series of experiments in different temperature and RH conditions, also combining an emulsifier adjuvant with the zeolite.

The research is not original since several kinds of similar materials have been tested and some became commercial products. However, the tested material seems to have peculiar features and the outcomes obtained are promising, so that, after being confirmed, hopefully, by field trials they might be directly applied to the control of the olive fruit fly.

The lab work is quite rigorous, although carried out with a limited number of insects. I explained my concern about some procedures in the attached file.

The manuscript is not perfectly balanced: in the introduction, the part dealing with features of zeolites is a bit too long and, in my opinion, some articles published by authors of the same group should be omitted without impairing the manuscript. On the contrary, the discussion chapter does not consider important aspects such as a more extensive comparison with similar materials used in IPM (kaolin and other clays) or with products showing deterrent effects (copper compounds). I think this part should be improved.

In the introduction and discussion chapter many citations are not properly cited or not relevant at all. Also, there are many flaws and mistakes in the reference list.

 Further comments and suggestions are detailed on a point to point basis in the attached file.

Comments on the Quality of English Language

In general, the English language of the manuscript is fine. However, although I am not English mother-tongue I suggest to edit or rephrase some sentence, as specified in the attached file.
